

**The Potential of Integrating Landscape, Geochemical and**
**Economical Indices to Analyze Watershed Ecological Environment**
**Huan Yu [a, d], Bo Kong [b], Zheng-Wei He [a], Guangxing Wang [c], Qing Wang [c]**
[a] *College of Earth Sciences, Chengdu University of Technology, 610059, Chengdu, China*
[b] *Institute of Mountain Hazards and Environment, Chinese Academy of Sciences, 610041, Chengdu, China*
[c] *Department of Geography and Environmental Resources, Southern Illinois University, 62901, Carbondale, USA*
[d] *Key Laboratory of Geoscience Spatial Information Technology of Ministry of Land and Resources, Chengdu*
*University of Technology, China*
Correspondence should be addressed to Huan Yu, Email: yuhuan0622@126.com; Telephone: 8618602846902;
Fax: 8602884075175.
**Abstract:**
A river watershed is a complicated ecosystem, and its spatial structure and
temporal dynamics are driven by various natural factors such as soil properties and
topographic features, human activities, and their interactions. Thus, characterizing the
river watershed ecosystem and monitoring its dynamics is very challenging. In this
study, we explored the characteristics of the ecosystem and environment of Yalong
River watershed in Ganzi Tibetan Autonomous Prefecture, Sichuan Province of China
by analyzing and modeling the relationships among economic indices, heavy metal
elements and landscape metrics. Landsat 8 data were used to generate a land cover
classification map and to derive landscape pattern indices. Governmental finance
statistics yearbook data were referred to provide economic indices. Moreover, a total
of 9 water samples were collected from the upstream to the downstream to obtain the
values of heavy metal concentrations in the water body. Then, both correlation and
regression analyses were applied to analyze and model the relationships among these
indices. The results of this study showed that 1) The ecological status and process of
this river watershed could be explained by analyzing the relationships among the



economic indices, heavy metal elements and landscape pattern indices selected based
on correlation analysis; 2) Compared with the economic indices, the accumulated
economic indices were more significantly correlated with most of the heavy metal
elements and should be applied for the integrated assessment of the watershed
ecological environment; 3) Landscape pattern indices SHDI and IJI had strong
correlations with the important economic indices Population and Population Density
and could be used for the integrated assessment of the watershed characteristics; 4)
Compared with land cover area, land cover area ratios were more sensitive to the
variation of the economic indices. The dominated land cover types Forest and
Grassland had strong relationships with the economic indices; and 5) Cu and Zn had
significant correlations with the landscape pattern indices. This study implied that
analyzing and modeling the relationships among the economic indices, heavy metal
elements and landscape pattern indices can provide a powerful tool for characterizing
the ecosystem of the river watershed and useful guidelines for the watershed
management and sustainable development.
**Keywords**:Watershed; Geochemistry; Landscape; economic indices; remote sensing;
Statistical analysis.

## 1. Introduction

The challenge of balancing human needs for water with environmental
sustainability has come to a head in river systems, where various management plans
to conserve and manage the ecosystems have been thrown into a turmoil (Pincock
2010). River ecosystems are mainly influenced by integrated biological, chemical and
physical subsystems, which increases uncertainty in ecological assessments, and
hampers prediction for the ecological environment changes (Wiley et al. 2010). An




in-depth understanding of ecological status and process in river systems is very
important for river conservation and management (Wang and Yang 2014). Stream
flow and water quality of a river are affected by both natural and anthropogenic
factors that exist within a watershed, hence the watershed has been recognized as an
appropriate analysis unit for addressing the challenges of water management (Singh et
al. 2014; Deng et al. 2014).
There is an urgent demand for sustaining or improving the functions of
watersheds to strengthen their roles in supporting human and meeting ecosystem
needs simultaneously, because watersheds provide economic goods and ecological
services that impact the livelihoods of people (Ingram et al. 2012). Benefiting the
economy, community and environment synchronously would realize the sustainable
development of a watershed. To achieve this goal, a proactive approach that combines
information of economic, social and ecological influence is needed (Randhir and
Shriver 2009; Kantamaneni 2016). Thus, the opportunity for sustaining human and
their river systems can be enhanced by examining how socioeconomic and ecological
processes are integrated at the watershed level (Wolters and Kuenzer 2015; Naiman

1992).

Human activity induced disturbances are one of the most important factors that
generate potentially permanent changes to the ecological structure and functions of
watersheds (Wang et al. 2015). The pattern and process of land use (or land cover) is
one typical manifestation of the interaction between human activities and ecological
processes observed in a region (Naiman 1992). Both the extent and depth of
transformation are determined by regional land use patterns and processes (Kabat et al.
2004). Understanding how human depends on landscape functions and products, and
how land use affects ecological and socioeconomic processes can provide a sound



basis for guiding sustainable development of a watershed (Naveh and Lieberman
1984; Zonneveld and Forman 1990).
Landscape ecology is a subsidiary discipline of modern ecology, which deals
with the interrelationship between human and landscapes that they live on (Naveh and
Lieberman 1990). Landscape ecology focuses on the interactions between landscape
patterns and ecological processes, and exploring the impacts of land use patterns on
water quality and the spatial scales over which these effects are manifest has become
a significant theme of landscape ecological studies (Turner et al. 2001). Digitized land
use data stored in a Geographical Information System (GIS) are always used to
conduct the analysis of landscape patterns, especially, landscape-level ecosystem
status can be credibly estimated through landscape measurements based on land use
data obtained from remote sensing imagery (Johnson and Patil 2006). However, as the
landscape patterns and ecological processes interact in diverse ways, neither of them
can be ignored to grasp the synthetical dynamics of the environment (Fu and Jones
2013). Although various factors including social, economic, and ecological
considerations that interactively determine landscape patterns are known abstractly,
the quantitative interrelationships among these variables are inadequately recognized.
Furthermore, it is difficult to describe the behaviors of a landscape scaling up from
ecological systems to communities, thus in-depth exploration of the relationships
between landscape patterns and ecological processes is necessary. Because a
landscape presents macroscopic and vast scale characters, which cannot be described
and studied at a microscopic level, the landscape and geochemistry interaction will be
a crucial challenge for studying on the ecological environment assessment in the
coming decades. For example, many recent studies have focused on the influence of
land use patterns in watersheds on water quality and biological communities in



streams (Vrebos et al. 2017; Vaighan et al. 2017; Dzinomwa and Ndagurwa 2017).

Geochemistry is the study of the distribution and migration of elements in the

environment where we live in, aiming at exploring the distribution of elements in the
earth and interpreting the processes that induce these distribution patterns based on
techniques and principles of chemistry and physics (Wainerdi and Uken 1971).
Hydro-geochemical investigations of surface water can provide information on the
extent and degree of element impacts so as to estimate the level of pollution and
identify principal pollutants in surface water (Quercia and Vidojevic 2012).
Hydro-geochemical speciation methods can offer a more realistic and reliable
measure to identify the degree of migrated water contamination, because they provide
fundamental ideas for better understanding of water features and they have a
sophisticated and meticulous methodology (Moldan 1992; Reuther 1996). However,
the transportation of particulate and dissolved materials in river systems is a
complicated action of different biological, chemical and physical processes occurring
in the watersheds and in the water (Hedges et al. 1986). Hence, available information
on trace elements, including heavy metals in water, is generally inadequate for
regional studies of the ecological environment, and little systematic information on
the spatial relationships between geochemistry and ecology of water is available
(Bowie and Thornton 1985). Thus, a fundamental question concerns whether we can
detect, describe and predict the ecological effects at the geochemical level has been
proposed (Reuther et al. 1996). Then, applying landscape and geochemistry integrated
methods to analyze the ecological environment of a watershed has its theoretical basis
and practical need. Furthermore, socioeconomic and ecological processes need to be
combined to obtain a sustainable development of a watershed at the landscape level,
on which all kinds of analyses utilize land cover types as the basic unit of calculation.



First of all, landscape pattern indices, characterizing diversified aspects of
composition, structure and spatial configuration of landscapes, were introduced to
quantitatively describe the correlations between spatial patterns and ecological
processes (O'Neill et al. 1988; Remmel and Csillag 2003). One of the most
fascinating features of landscape pattern indices is the simplicity: large amount of data
can be summarized by a single number (or by a limited set of numbers) without a
priori knowledge about the processes and organisms of landscapes (Fortin et al. 2003).
Besides, heavy metals are especially dangerous elements and expose potential
ecological risks to living organisms, on account of their bioaccumulation,
non-degradability and toxicity features (Cai et al. 2015). Heavy metal contamination
in aquatic ecosystems is frequently surveyed by evaluating concentrations in
sediments, biota and water (Rahman et al. 2014), in which variations of the heavy
metal distributions can provide direct information for evaluating the status of
pollution and baseline data to help further develop an efficient strategy on their
controls (Dong et al. 2015; Yeh et al. 1977). Apart from that, economic indices
provide supplementary information on the strength of human activities that give rise
to the production of pollutants (Zhou et al. 2012). For example, the Gross Domestic
Product (GDP) is commonly used as an index for evaluating the economic health and
measuring the living standard of a country. Because the sustainable development of
watersheds requires an integration of hydrologic, ecological and socio-economic
aspects, relationships among these indices or indicators involving landscape pattern,
geochemistry and economy need to be explored to gain an in-depth understanding of
ecological processes and properties in a watershed.
The main goals of this study are to: (a) analyze whether and how the
relationships among these indices including landscape pattern, geochemistry and



economy can be found, and (b) explore the potential of analyzing the ecological
environment of a watershed based on a landscape, geochemistry and economy
integrated view.
**2. Materials and Methods**
**2.1 Study area and sampling**
The study area was Yalong River watershed, within Ganzi Tibetan Autonomous
Prefecture, Sichuan Province (Fig. 1). The study area has a total area of 70,366 km$^2$,
and 3/5$^{th}$ of the Yalong River's full length is distributed in the study area. This region
is located in the upstream section of the Yalong River, which has an important
influence on the water quality and ecological environment. Covering a total of six
counties including Shiqu, Dege, Ganzi, Xinlong, Litang and Yajiang in the
administrative regions of Ganzi, most of the area is mountainous with steep terrain. In
Shiqu County located in the upstream of the basin, the average elevation is 4526.9 m,
the average annual temperature is below -1.6 ℃, and the average annual precipitation
is 569.6 mm. However, in Yajiang County located in the downstream of the basin, the
lowest elevation is 2266 m, the average annual temperature is below 11 ℃, and the
average annual rainfall is 650 mm. The regional vertical variations of temperature,
precipitation, and vegetation are obvious with the terrain height changes (Shen et al.

2010; 2012).

In total, 9 water samples were collected in the study area in 2014 (Fig. 1). The
sampling locations were steadily scattered in the study area from its upstream to
downstream to survey heavy metal concentration characteristics in the water body. A
hand-held global positioning system (GPS) receiver was used to record the exact
locations of the samples for further being imported into ArcGIS. In order to perform



the parametric statistical analysis, 30 observation points were obtained through the
interpolation of 9 water samples. Furthermore, an identify function of ArcGIS was
used to acquire the data of landscape pattern and economy indices for statistical
analysis based on the 30 observation points.
**2.2 Measuring landscape pattern metrics**
2.2.1 Source of data

This study collected and used six multi-spectral bands (band 2-blue, band

3-green, band 4-red, band 5-near infrared, band 6-shortwave channel 1, and band
7-shortwave channel 2) of Landsat 8 images at the spatial resolution of 30 m × 30 m
to classifying land cover types of this study area and obtain land cover maps. A total
of nine cloud-free leave-on and leave-off images dated from May of 2013 to Jan. of
2014 were acquired by downloading from the website supported by USGS (United
States Geological Survey). The radiometric correction and geometric correction of the
images were first conducted and then were clipped according to the boundary of the
study area with ENVI software.
2.2.2 Land cover classification

The establishment of a scientific land cover classification system according to

the regional condition is the primary work needed to obtain the regional landscape
data (Anderson et al. 1976; Li and Ma 2000; Bazi and Melgani 2006). Reference for
the classification system was made to the land use and land cover classification
system for remote sensing data from USGS, the national land classification (For
Transition Period) from Ministry of Land and Resources of P. R. China, as well as the
regional condition of land cover in the Yalong River watershed, and the requirements
for further study. The regional landscape was classified into: forest, river, grassland,



lake, marsh land, bare soil, farm land, human habitation, industrial land, glacial and
snow, and transportation land.

Based on the Yalong River watershed land cover classification system, a strict

description for each type of land cover class was obtained. An object-oriented
classification method was applied to extract the land cover information of the study
area. Unlike the traditional classification methods that analyze spectral information of
land cover types, the object-oriented classification method accounts for the spatial
characteristics such as shape and compactness of objects and the relationships
between the objects (Sapozhnikova et al. 2006; Kassouk et al. 2014). This method
first carried out multi-scale image segmentation, that is, classified the pixels into
homogeneous polygons (objects) based on their similarity measured using variances
of pixel values, and shape, smoothness and compactness of objects. The classification
of land cover types was then conducted using decision tree and nearest neighbor. In
order to improve the accuracy of the classification, expert knowledge was applied to
conduct the verification and interpretation.
2.2.3 Obtaining land cover map

Based on the above methods, this study obtained the land cover map of the

Yalong River watershed (Fig. 2). In accordance with the statistics of the classification
results, the areas and proportions of the land cover types of the landscape were
obtained, as shown in the Table 1. The statistics showed that the grassland and forest
were the major land cover types with their area accounting for 89% of the entire
watershed. As shown in Fig. 2, the land cover map of the Yalong River was smooth
and compact due to the segmentation of the objects, without the traditional 'salt and
pepper' phenomenon formed by isolated pixels. Also, the segments contained
information such as shapes, veins, space, and so on, which could be comprehensively



utilized in the process of the classification.

In this study, a 30 m spatial resolution image was used in the classification. To

ensure the results of the classification accuracy assessment were objective, the
samples used for the accuracy assessment were selected from the 1 m spatial
resolution image provided by Google Earth. A total of 450 samples were obtained by
a simple random sampling method, and these samples were used to calculate the
confusion matrix (Foody 2002). The overall accuracy of the classification was 87.11%,
and the Kappa coefficient was 0.855. Therefore, the high accuracy could fully meet
the demand of this study.
2.2.4 Computing landscape pattern metrics

Landscape pattern indices are easy to understand due to their ecological

meanings. The indices also contain certain statistic characteristics and are easily used
to analyze and compare the sizes of different patches, and provide important
information of landscape patterns, structures and spatial composition to explain the
functions of landscapes. Landscape pattern indices have been widely used to describe
landscape patterns and changes, and to set up the contact between the patterns and
landscape processes (Turner et al. 2001).

Considering the aims of this study and the features of every landscape pattern

index, the follow indices were chosen as the indicators to quantify the ecological
features: Total Area (TA) represents the area of each landscape type; Total Edge (TE)
equals to the sum of the edge lengths of all the segments involved in a corresponding
patch type; Edge Density (ED) means the sum of the edge lengths of all segments
involving a corresponding patch type and divided by the total landscape area;
Contagion (CONTAG) is the negative sum of the proportional abundance of each
patch type and multiplied by the proportion of the adjacencies between the cells of



this patch type and another patch type; Percentage of Like Adjacencies (PLADJ) is
computed as the sum of the diagonal elements of the adjacency matrix and divided by
the total number of adjacencies; Interspersion & Juxtaposition Index (IJI) considers
all the patch types present on an image to analyze the amount of patch adjacency or
fragmentation; Patch Cohesion Index (COHESION) is computed from the
information contained in the patch area and the perimeter; Landscape Division Index
(DIVISION) is defined as the probability that two animals placed within different
areas somewhere in the region of the investigation might find each other; Effective
Mesh Size (MESH) simply denotes the size of the patches when the landscape is
divided into S areas, with the same degree of landscape division as obtained for the
observed cumulative area distribution; Splitting Index (SPLIT) is defined as the
number of patches obtained when the total landscape is divided into the patches of
equal size, in such a way that this new configuration leads to the same degree of
landscape division as obtained for the observed cumulative area distribution;
Shannon's Diversity Index (SHDI) is the representative of diversity of a landscape;
Number of Patches (NP) reflects the number of all the landscape patch types; Patch
Density (PD) measures the heterogeneity of the landscape; Largest Patch Index (LPI)
indicates the influencing extent of the largest plaque for the entire landscape;
Landscape Shape Index (LSI) reflects the divergence of the shape of landscape
patches from the ideal circle; and Aggregation Index (AI) means the percentage of
like adjacencies between cells of same patch type (McGarigal et al. 2012).
**2.3 Measuring chemical concentration**
The chemical parameters (Al, Fe, Cr, Ni, Cu, Zn, Cd, Pb) were measured
according to the industry standard (DZ/T0064-93) (Figs. 3-6), conducted by Ministry
of Geology and Mineral Resources of P. R. China. All these elements were measured





with a method of ICP-MS (Inductively Coupled Plasma Mass Spectrometry). The
results are listed in Table 3.
**2.4 Measuring economic variables**
The GDP and Population are commonly used as the indicators for measuring the
economic health and living standard in a country. In addition to these two indices,
other indices are also used to observe the effects of human disturbances on water
quality. Data pertaining to spatial distribution of economic indices (Population,
Accumulated Population, Population Density, Accumulated Population Density, GDP,
Accumulated GDP, Per Capita GDP, Accumulated Per Capita GDP, Gross Output
Value of Agriculture, and Accumulated Gross Output Value of Agriculture) were
generated through the spatial analysis methods, to identify the relationships between
economic indicators and other indices (Table 2). As examples, Fig.7 and Fig. 8
respectively showed the spatial distributions of GDP and its accumulation values. The
accumulated indicators were calculated by summing the local values of the
corresponding indicator along the river from the upper reach and implied the impacts
of accumulated values.
**2.5 Statistical analysis**
The relationships between ecological, hydrologic and socio-economic factors,
respectively, were analyzed by calculating Pearson product moment correlation
coefficients among the landscape pattern, geochemistry and economy indices
(Pearson 1895). Moreover, a significance test was conducted to determine whether the
coefficients were statistically significantly different from zero at the significant level
of 0.05. Linear and nonlinear relationships between the economic indices, heavy
metal elements and landscape pattern indices were then tested and quantitatively



explored through various linear and non-linear regression models. In addition to linear
models, the potential nonlinear equations include exponential, growth, logistic,
S-curve, compound, power, cubic, quadratic, inverse, and logarithmic models. The
most appropriate models were obtained through the comprehensive analysis of
coefficient of determination ($R^2$) and statistical significance (Sig.). Finally, principal
component analysis (PCA) is a widely used method to reduce the dimensions of
variables. In this study, PCA was carried out to reduce the number of the original
variables (Polit and Beck 2012). At the same time, PCA was also used to identify the
interrelationships among the landscape pattern, heavy metal elements and economic
indices, and to determine whether and how the relationships among them could be
presented by specific representative factors.
**3. Results and discussion**
**3.1 Distribution characteristics of elements in water samples**

Figures 3-6 show the values of water quality parameters for the upper, middle,

and lower main channels. The contents of Cr, Ni, Cu, Zn, Cd, and Pb were all below
the guideline values for Drinking-water Quality defined by World Health
Organization and the Environmental Quality Standards for Surface Water by the
Ministry of Environmental Protection of P. R. China. However, the contents of Al and
Fe were significantly higher than the guideline values. Spatially, the contents of the
elements in the river water generally increased from the upper to downstream. Table 3
summarized the mean values of the water quality parameters for counties from the
upstream to the downstream. The average values of Al, Fe, Ni, Zn and Pb
continuously increased as the water flew to the downstream. The spatial pattern is
somewhat alike to that gained in the study of the Fuji River in Japan, in which





high-pollution regions were mainly located in the downstream (Shrestha and Kazama
2007). However, the spatial distributions of Cr, Cu, and Cd values fluctuated from the
upstream to the downstream.
**3.2 Analysis of landscape pattern**
Based on the land cover classification results, the landscape pattern indices of the
study area were obtained using Fragstats 4.2 software, which are shown in Table 4.
The results indicated that the values of the indices were diversified from the upstream
to the downstream except PLADJ, AI, COHESION and MESH. Among the indices,
TA, LPI, CONTAG, PLADJ, COHESION, MESH and AI showed the highest values,
while PD, ED, LSI, DIVISION, SPLIT and SHDI had the lowest values in Shiqu
county located in the upstream, implying that a health ecological condition was
observed in the upstream. In Xinlong county located in the midstream, there were
highest values for NP, PD, TE, ED, LSI, DIVISION, SPLIT and SHDI, and lowest
values for LPI, PLADJ, IJI and AI, demonstrating that the ecological environment
was disturbed and landscape fragmentation was observed. The landscape indices LPI,
PD and DIVISION showed a turning point in the midstream Xinlong County. In
Litang county that had a smallest area, the values of TA, NP, TE, CONTAG,
COHESION and MESH were lowest, while the value of IJI was highest, indicating
that the ecological environment needed to be paid attention to.
The AI in all the counties had the values of above 91.5, indicating that the
landscape of the study area showed a high degree of aggregation, that is, the
ecological environment was still in a good condition. The differences of LPI between
the counties were very obvious. All the high values were distributed in the upstream,
which meant the large patches dominated the landscape of the region. LSI had the
higher values in the downstream, which indicated that the landscape structure was

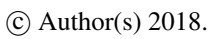



complicated in this region. In addition, by combining the values of CONTAG, PLADJ,
COHESION, DIVISION, MESH, SPLIT and SHDI indices in the table, it was found
that the upstream had a better but weaker ecological condition than the downstream.
**3.3 Correlations among landscape pattern, geochemistry and economy indices**
3.3.1 Economic indices and heavy metal elements

The Pearson correlation coefficients between the economic indices and heavy

metal elements were calculated. The Al, Fe, Ni and Pb elements were significantly
correlated with the economic indices at the significant level of 0.05 except Population,
Population Density and Gross Output Value of Agriculture. The Cr had weak
relationships with the economic indices except GDP. The Cu element had a significant
correlations with the economic indices except Population, Population Density, Per
Capita GDP and Gross Output Value of Agriculture. The Zn was significantly
correlated with the economic indices except Population Density, GDP and Gross
Output Value of Agriculture. There were significant correlations between Cd element
and the economic indices except Population Density, Accumulated Population Density,
Per Capita GDP and Gross Output Value of Agriculture. In summary, GDP and its
relevant indices showed significant correlations with most of the heavy metal
elements; and all the accumulated indices were also significantly correlated with most
of the heavy metal elements because of the assembling characteristics of elements
from the upstream to the downstream.
3.3.2 Economic and landscape pattern indices

The correlation coefficients between the economic indices and landscape pattern

indices were calculated. PD is a fundamental manifestation of landscape patterns and
can provide the information on the degree of landscape fragmentation. A significantly



negative correlation between PD and Population looked like inexplicable and
controversial because an increased population potentially resulted in landscape
fragmentation. The limitation of PD index emerged when it was used for expressing
the number of patches to assist the comparisons among the varying size landscapes.
SHDI is a popular measure of diversity in ecology. In this study, it had a significantly
negative correlation with Population, which is very intelligible because the human
activities have negative influence on the ecological environment. TE is a gauge of the
total edge length of a specific patch type. Its limitation was also observed because it
showed the same distribution features with PD. IJI is based on patch adjacencies and
provides a measure of the interspersion or intermixing of patch types. In this study,
there was a significant correlation between IJI and Population Density, which is
understandable because a high intensity of human disturbances makes the regional
landscape structure more complicated. All these relationships proved that population
was complex and integrated with many social activities, which affected the landscape
patterns and structures. At the same time, some limitations of the landscape pattern
indices were also observed.
3.3.3 Economic indices and land cover area/area ratios
The Pearson correlation coefficients between the economic indices and land
cover area/area ratios showed that three main land cover types: Forest, Grassland and
Transportation Land had significant correlations with the economic indices, and land
cover area proportions or ratios were more sensitive to the variation of economic
indices. Transportation Land was highly correlated with the economic index
Population because the traffic condition was one key factor for supporting the local
human activities. Forest had a significantly negative correlation with Population,
which is intelligible because of the demand for woods and other indirect impacts on





the forest ecological system by human activities. However, GDP and Gross Output
Value of Agriculture, usually applied as the indicators of the economic health and
living standard in a country, almost do not have any relationships with the land cover
types, which seems inexplicable and controversial. This implies that a connection
between economic development and landscape structure is unable to be established
with a simple index.
3.3.4 Heavy metal elements and landscape pattern indices

The correlation coefficients of both Cu and Zn with CONTAG were -0.82 and

-0.84, respectively, showing significant negative correlations at the significant level of
0.05. CONTAG is influenced by both the dispersion and interspersion of patch types
and its negative correlation with the heavy metal element Zn is understandable
because one disaggregated and interspersed landscape pattern may release more Zn.
Both PD and DIVISION can provide fragmentation information for a regional
landscape. The significantly positive correlations of 0.82 were found between Zn and
PD and DIVISION, respectively. This is intelligible as the same reason with
CONTAG. However, no significant correlations were observed between all the other
heavy metal elements and landscape pattern indices, which indicates the limitation of
analyzing the relationships between the heavy metal elements and the landscape
pattern indices to explore the ecological status and process, and a more complicated
and comprehensive approach is needed.
3.3.5 Heavy metal elements and land cover area/area ratios

The Pearson correlation coefficients between the land cover area/area

proportions and the heavy metal elements showed that land cover area proportions
were more sensitive to the variations of the heavy metal elements than the land cover

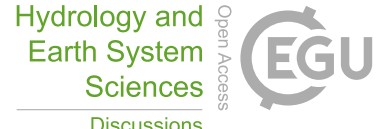

areas. Forest showed significantly positive correlations with all the heavy metals,
except Cr. Grassland showed strongly negative correlations with all the heavy metals
except Cr although the correlations were not statistically significant at the significant
level of 0.05. The moderately negative correlations between Transportation Land and
all the heavy metals except Cr and Cd could also be observed. River showed negative
correlations with the heavy metals. No strong correlations were observed between
other heavy metal elements and the land cover area proportions. The negative
correlations between the river and the heavy metals can be explained by a vast water
body that has a powerful ability to dilute the elements. However, the strong positive
correlations between forest and heavy metals, especially, the strong negative
correlations between transportation land and heavy metals are inexplicable and
controversial. There must be some profound causes for this paradoxical phenomenon.
For example, the features of the natural background were the main factors that control
the immigration process of geochemistry. These kinds of problems could not be
simply accounted for by analyzing the relationships between the heavy metal
elements and the land cover area/area proportions.

Overall, the above results showed that the regional ecological status and process

of the river watershed could be, to a certain extent, explained using the correlation
analysis among the hydrologic, ecological and socio-economic indices. However, the
ecological problems of the watershed are complex due to the interactions of the
hydrologic, ecological and socio-economic factors and could not be accounted for
through the simple correlation analysis. The sustainable development of the watershed
requires a comprehensive evaluation of the factors to gain an in-depth understanding
of the ecological processes and properties in the watershed.
3.3.6 The functional relationships between the indices



The functional relationships among the hydrologic, ecological and
socio-economic indices were analyzed using the indices that had statistically
significant correlations (Table 5). The results showed that the functional relationships
were diverse. It was found that as the indicators of the economic health in a country
and the references for quantifying the intensity of human activities, the economic
indices had significantly linear or nonlinear relationships with the heavy metal
elements and landscape pattern indices at the significant level of 0.05. The
accumulated economic indices such as accumulated population, accumulated
population density, accumulated GDP, etc., were more involved in the regression
models that were statistically significant than the economic indices themselves. Being
a sensitive indicator for water pollution, the variation of the heavy metal
concentrations in the river water was linearly or nonlinearly dependent on the
economic and landscape pattern indices.
The relationships of Forest, Grassland and Transportation Land with the heavy
metals were strong. Especially, the grassland can be considered as an intermediate and
disturbance dependent ecosystem (Pretelli et al. 2015) and provide various ecological
services, including soil and water conservation, carbon storage, and habitats for
animals and recreation (Sivanpillai and Shroder 2016). The grassland also plays an
important role in controlling the atmospheric greenhouse gases through carbon
storage and sequestration (O'Mara 2012; Gang et al. 2016). In addition, many studies
have proved that the correlations exist between grassland and heavy metals in diverse
ways or at different scales (Babalonas et al. 1997; Klessa and Desira-Buttigieg 1992;
Aitken 1997). As the dominant landscape type in this region, grassland accounted for
61.22 % of the study area, showing a strong correlation between the grassland
landscape and the heavy metal elements is intelligible and understandable.



Cu and Zn are crucial elements for both animals and plants, but they have also
been identified as possible specific pollutants in many countries (Comber et al. 2008;
Jensen et al. 2016). Cu and Zn account for the highest inputs of the trace elements in
agricultural soils (Tella et al. 2016), and many studies have proved that the
distribution patterns of Cu and Zn have significant correlations with certain landscape
patterns and processes (Stone and Droppo 1996; Lindström 2001; Morse et al. 2016).
This was supported by the results of this study.
All the heavy metals, landscape pattern and socio-economic indices listed in
Table 5 were selected to calculate the principal components of PCA (Table 6). The
results showed that there were three main components, and the top two accounted for
above 88% of the original variances, indicating principal component one and two
were enough for explaining the variability of the original and correlated variables.
Principal component one had large correlation coefficients with all the indices except
Population Density and IJI that showed great correlation coefficients with principal
component two. The results indicated that the interrelationships among the landscape
pattern, geochemistry and economy indices existed and were diverse, and could be
identified by the selected and representative factors. Moreover, some of the indices
involving in the analysis were important and could not be neglected, but should be
selected and integrated to analyze the characteristics of the ecological environment of
the river watershed.
**4. Conclusions**
In this study, the relationships between the economic indices, heavy metal
elements and landscape pattern indices were explored and used to analyze and
characterize the ecosystem and environment of Yalong River watershed within Ganzi
Tibetan Autonomous Prefecture, Sichuan Province, using water samples collected in





the field and an image derived land cover classification. In summary, this study led to
following findings: 1) The ecological status and process of the watershed could be
explained by analyzing the relationships among the economic indices, heavy metal
elements and landscape pattern indices selected based on correlation analysis; 2) The
accumulated economic indices than the economic indices themselves were more
significantly correlated with most of the heavy metal elements and should be applied
to the integrated assessment of the watershed ecological environment. This conforms
to the assembling characteristics of the elements in the river from the upstream to the
downstream (Cortecci et al. 2009; Li and Zhang 2010; Taylor et al. 2012; Yang et al.
2014; Bu et al. 2016); 3) Some landscape patterns indices such as SHDI and IJI could
be used to the integrated assessment of the watershed ecological environment because
of their strong correlations with the important economic indices, i.e., Population and
Population Density; however, some limitations of using the landscape pattern indices
were also observed, indicating that the selection of the landscape pattern indices was
essential; 4) Compared with land cover area, land cover area proportions were more
sensitive to the variation of the economic indices. As the dominated land cover types
in the study area, Forest and Grassland had strong relationships with the economic
indices. Transportation Land also had a close relationship with Population because
transportation and mobility were vital constituents of socio-economic development in
any country (Gentile and Noekel 2016; Alam et al. 2016); and 5) Cu and Zn were the
main elements that showed significant correlations with the landscape pattern indices
and this was also supported by previous studies (Stone and Droppo 1996; Lindström
2001; Morse et al. 2016). The conclusions will play a fundamental role in establishing
the synthetic models for management of watersheds.

However, it was found that the connection between the economic development





and the landscape structure was unable to be established with a simple index analysis.
Moreover, analyzing the relationships between the heavy metal elements and the
landscape pattern indices to explore the ecological status and process had their
limitations. The ecological problems of a watershed could not be revealed through
simply analyzing one kind of indices and the sustainable development of the
watershed requires an integrated evaluation of hydrologic, ecological and
socio-economic factors. A more complicated and comprehensive approach is needed
to get an in-depth understanding of the ecological processes and properties of the
watershed. Although, at the present, an increasing number of theories and methods for
integrated watershed management have been developed, the exploration of
quantitative relationships among the driving factors still requires a significant effort
and multivariate statistical methods based on sufficient sampling data in the future
work could be an alternative.

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




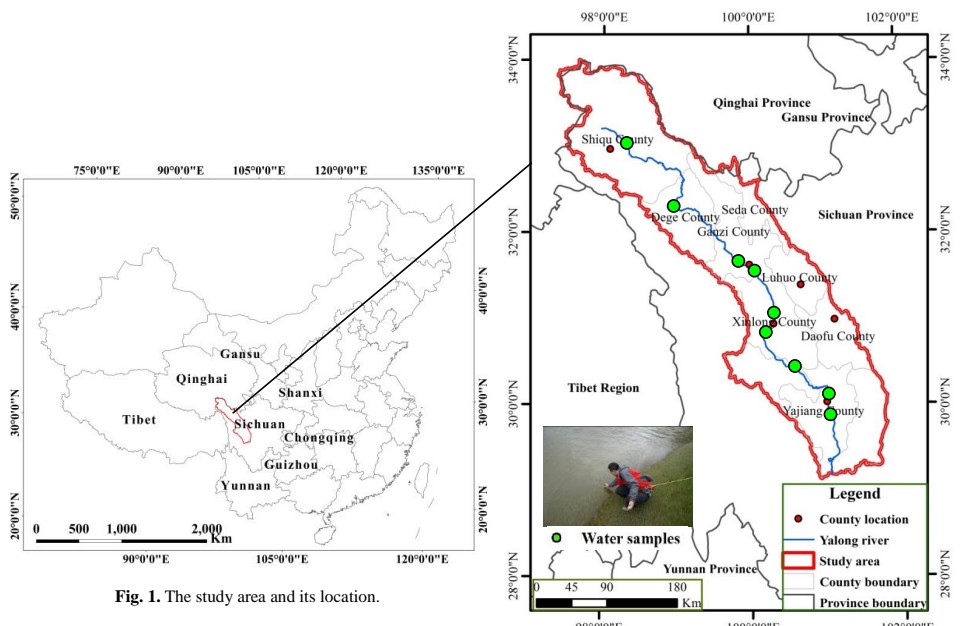

**Fig. 1.** The study area and its location.

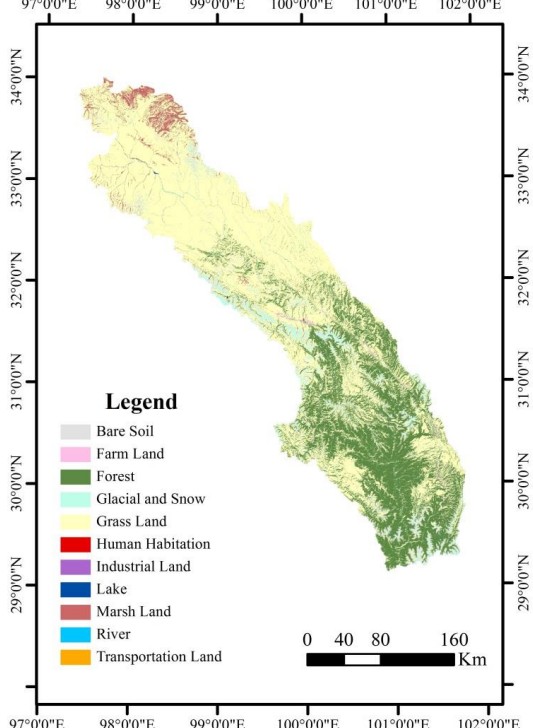

**Fig. 2.** Land cover classification map.




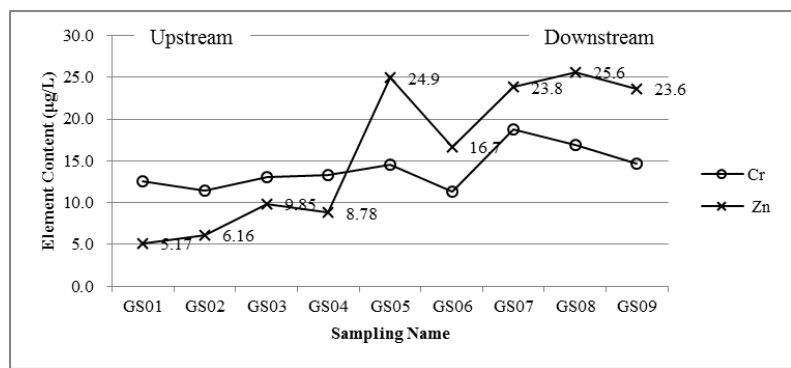

**Fig. 3.** The values of Cr and Zn elements from water samples along the river from

upstream to downstream (μg/L)

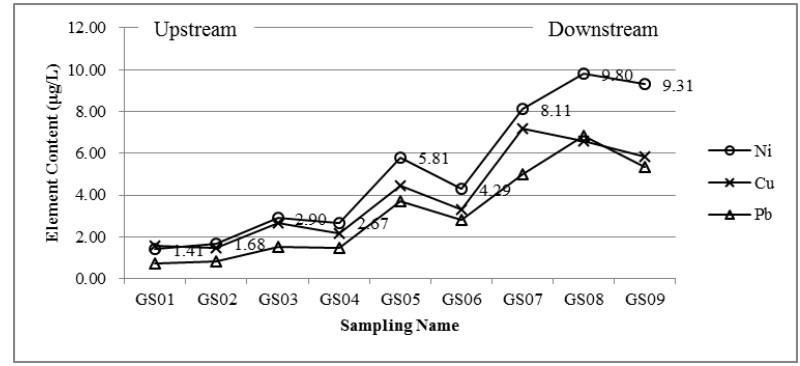

**Fig. 4.** The values of Ni, Cu and Pb elements from water samples along the river from

upstream to downstream (μg/L)

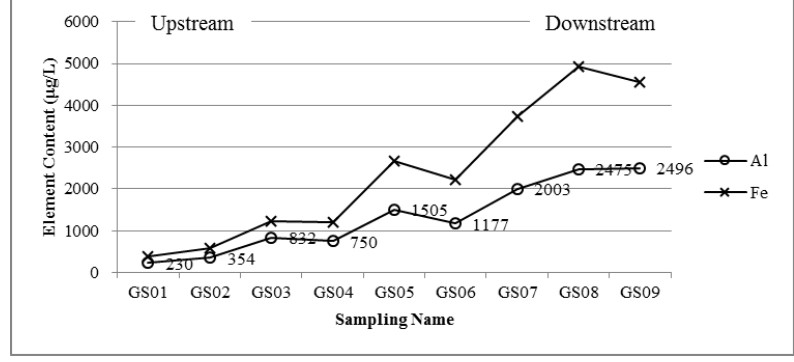

**Fig. 5.** The values of Al and Fe elements from water samples along the river from

upstream to downstream (μg/L)





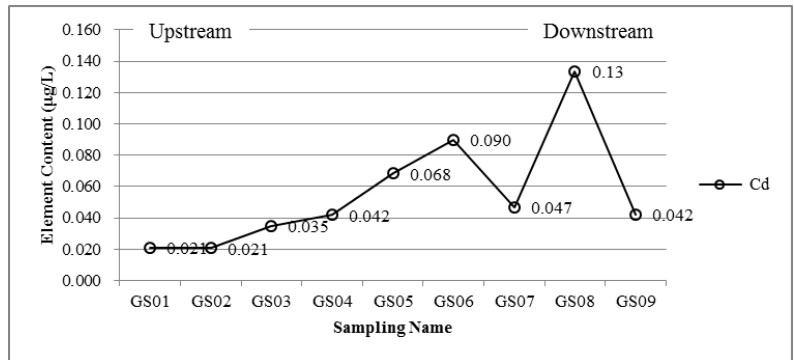

**Fig. 6.** The values of Cd element from water samples along the river from upstream to

downstream (µg/L)

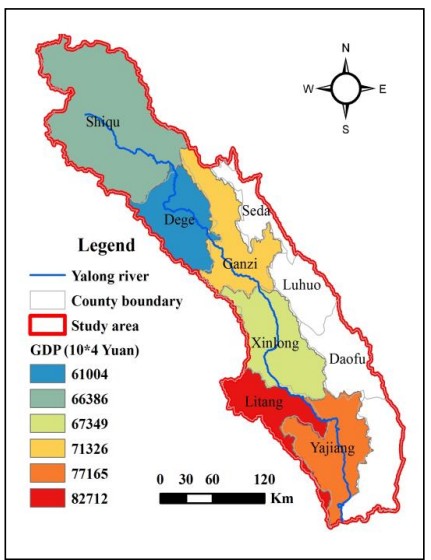

**Fig. 7.** Map showing the GDP of the counties that
the river goes through

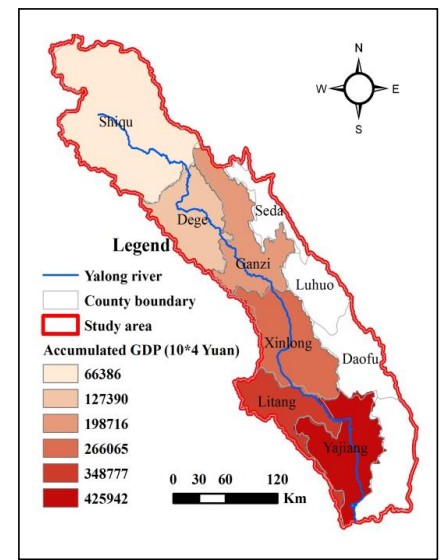

**Fig. 8.** Map showing the accumulated GDP of the
counties that the river goes through





**Table 1.** Area and proportion of each land cover type.

| Land cover classes | Shiqu | | Dege | | Ganzi | |
|---|---|---|---|---|---|---|
| | Area(km$^2$) | Proportion (%) | Area(km$^2$) | Proportion (%) | Area(km$^2$) | Proportion (%) |
| Bare Soil | 561.77 | 3.12% | 398.24 | 6.18% | 309.39 | 4.22% |
| Farm Land | 4.81 | 0.03% | 22.86 | 0.35% | 80.18 | 1.09% |
| Forest | 1.10 | 0.01% | 644.45 | 10.00% | 774.57 | 10.56% |
| Glacial and Snow | 131.10 | 0.73% | 286.03 | 4.44% | 285.20 | 3.89% |
| Grassland | 15428.98 | 85.68% | 4997.31 | 77.57% | 5793.54 | 78.96% |
| Human Habitation | 1.45 | 0.01% | 2.65 | 0.04% | 7.66 | 0.10% |
| Industrial Land | 0.45 | 0.00% | 0.23 | 0.00% | 0.19 | 0.00% |
| Lake | 9.48 | 0.05% | 5.18 | 0.08% | 0.19 | 0.00% |
| Marsh Land | 1693.74 | 9.41% | 36.08 | 0.56% | 23.84 | 0.32% |
| River | 143.26 | 0.80% | 38.94 | 0.60% | 55.63 | 0.76% |
| Transportation Land | 32.26 | 0.18% | 10.68 | 0.17% | 6.86 | 0.09% |

**Table 1 (Continuous).** Area and proportion of each land cover type

| Land cover classes | Xinlong | | Litang | | Yajiang | |
|---|---|---|---|---|---|---|
| | Area(km$^2$) | Proportion (%) | Area(km$^2$) | Proportion (%) | Area(km$^2$) | Proportion (%) |
| Bare Soil | 451.32 | 5.64% | 191.70 | 4.11% | 363.59 | 4.80% |
| Farm Land | 38.69 | 0.48% | 15.72 | 0.34% | 38.01 | 0.50% |
| Forest | 3540.41 | 44.22% | 2274.36 | 48.78% | 4868.50 | 64.29% |
| Glacial and Snow | 330.76 | 4.13% | 104.38 | 2.24% | 81.37 | 1.07% |
| Grassland | 3581.73 | 44.74% | 2062.01 | 44.23% | 2181.47 | 28.81% |
| Human Habitation | 1.00 | 0.01% | 0.00 | 0.00% | 0.98 | 0.01% |
| Industrial Land | 0.37 | 0.00% | 0.00 | 0.00% | 0.11 | 0.00% |
| Lake | 13.62 | 0.17% | 0.00 | 0.00% | 1.02 | 0.01% |
| Marsh Land | 5.77 | 0.07% | 0.00 | 0.00% | 0.00 | 0.00% |
| River | 39.59 | 0.49% | 14.16 | 0.30% | 37.88 | 0.50% |
| Transportation Land | 2.89 | 0.04% | 0 | 0.00% | 0 | 0.00% |


**Table 2.** Statistics of economic indices for counties along the river.

| Counties | Population (10⁴Persons) | Accumulated Population (10⁴Persons) | Population Density (Persons/km²) | Accumulated Population Density (Persons/km²) | GDP (10⁴Yuan) | Accumulated GDP (10⁴Yuan) | Per Capita GDP (Yuan) | Accumulated Per Capita GDP (Yuan) | Gross Output Value of Agriculture (10⁴Yuan) | Accumulated Gross Output Value of Agriculture (10⁴Yuan) |
|---|---|---|---|---|---|---|---|---|---|---|
| Shiqu | 9.77 | 9.77 | 4.18 | 4.18 | 66386 | 66386 | 6795 | 6795 | 45252 | 45252 |
| Dege | 8.26 | 18.03 | 12.13 | 16.21 | 61004 | 127390 | 7385 | 14180 | 36568 | 81820 |
| Ganzi | 6.94 | 24.97 | 8.17 | 24.38 | 71326 | 198716 | 10278 | 24458 | 44344 | 126164 |
| Xinlong | 5.12 | 30.09 | 5.66 | 30.04 | 67349 | 266065 | 13154 | 37612 | 36489 | 162653 |
| Litang | 7.01 | 37.10 | 12.6 | 42.64 | 82712 | 348777 | 11799 | 49411 | 39616 | 202269 |
| Yajiang | 5.12 | 42.22 | 6.13 | 48.77 | 77165 | 425942 | 15071 | 64482 | 27692 | 229961 |





**Table 3.** Statistics of heavy metal elements for counties along the river.

| Counties | Al | Cr | Fe | Ni | Cu | Zn | Cd | Pb |
|----------|----|----|----|----|----|----|----|----|
|          | μg/L | μg/L | μg/L | μg/L | μg/L | μg/L | μg/L | μg/L |
| Shiqu    | 230  | 12.5 | 377  | 1.41 | 1.57 | 5.17  | 0.021 | 0.74 |
| Dege     | 354  | 11.4 | 586  | 1.68 | 1.49 | 6.16  | 0.021 | 0.82 |
| Ganzi    | 791  | 13.2 | 1218 | 2.78 | 2.41 | 9.32  | 0.038 | 1.51 |
| Xinlong  | 1341 | 12.9 | 2447 | 5.05 | 3.89 | 20.80 | 0.079 | 3.28 |
| Litang   | 2003 | 18.8 | 3732 | 8.11 | 7.21 | 23.80 | 0.047 | 4.99 |
| Yajiang  | 2485 | 15.8 | 4742 | 9.56 | 6.20 | 24.60 | 0.088 | 6.11 |



**Table 4.** Landscape pattern indices of the study area.

| Region | TA | NP | PD | LPI | TE | ED | LSI | CONTAG |
|---|---|---|---|---|---|---|---|---|
| | PLADJ | LII | COHESION | DIVISION | MESH | SPLIT | SHDI | AI |
| Shiqu | 1800824 | 22492 | 1.25 | 82.39 | 38454870 | 21.35 | 73.64 | 85.28 |
| | 96.75 | 36.98 | 99.96 | 0.32 | 1223736.26 | 1.47 | 0.56 | 96.79 |
| Dege | 644263 | 17027 | 2.64 | 71.40 | 240016890 | 37.28 | 76.85 | 77.48 |
| | 94.33 | 41.11 | 99.91 | 0.49 | 328937.41 | 1.96 | 0.84 | 94.40 |
| Ganzi | 733726 | 19702 | 2.69 | 62.68 | 25446540 | 34.68 | 77.08 | 78.44 |
| | 94.70 | 42.94 | 99.89 | 0.59 | 298410.58 | 2.46 | 0.80 | 94.76 |
| Xinglong | 800609 | 32222 | 4.02 | 21.15 | 42916860 | 53.61 | 121.78 | 70.40 |
| | 91.90 | 35.34 | 99.81 | 0.90 | 79176.66 | 10.11 | 1.09 | 91.96 |
| Litang | 466235 | 16748 | 3.59 | 31.13 | 21566220 | 46.26 | 82.39 | 65.07 |
| | 92.91 | 46.34 | 99.77 | 0.85 | 69570.70 | 6.70 | 0.96 | 92.98 |
| Yajiang | 757294 | 24862 | 3.28 | 58.83 | 28723440 | 37.93 | 85.03 | 73.96 |
| | 94.22 | 37.34 | 99.89 | 0.64 | 270170.13 | 2.80 | 0.89 | 94.28 |




**Table 5.** The functional relationships between the economic indices, heavy metal elements and landscape pattern indices

| Indices A (y) | Indices B (x) | Functional Relationships | Indices A (y) | Indices B (x) | Functional Relationships |
|---|---|---|---|---|---|
| Population | Zn | $y = 4.608 + 24.537/x$, ($R^2=0.82$, Sig.$=0.013$) | Population | SHDI | $y = 5.868 - 6.669\ln(x)$, ($R^2=0.71$, Sig.$=0.035$) |
| Population | Cd | $y = 1.92*x^{-0.40}$, ($R^2=0.94$, Sig.$=0.002$) | Population | Forest | $y = 5.834 - 0.458\ln(x)$, ($R^2=0.71$, Sig.$=0.035$) |
| Accumulated Population | Al | $y = -57.216 + 12.449\ln(x)$, ($R^2=0.98$, Sig.$=0.000$) | Population | Grassland | $y = 3.923*2.529^x$, ($R^2=0.72$, Sig.$=0.032$) |
| Accumulated Population | Fe | $y = -58.167 + 11.644\ln(x)$, ($R^2=0.98$, Sig.$=0.000$) | Population | Transportation Land | $y = 5.528 + 1885.276x$, ($R^2=0.71$, Sig.$=0.035$) |
| Accumulated Population | Ni | $y = 7.845 + 14.654\ln(x)$, ($R^2=0.96$, Sig.$=0.001$) | Accumulated Population | Forest | $y = 14.057 + 43.765x$, ($R^2=0.90$, Sig.$=0.004$) |
| Accumulated Population | Cu | $y = 45.469 - 48.523/x$, ($R^2=0.89$, Sig.$=0.005$) | Accumulated Population | Grassland | $y = 55.491 - 47.437x$, ($R^2=0.86$, Sig.$=0.007$) |
| Accumulated Population | Zn | $y = \exp(3.957 - 7.731/x)$, ($R^2=0.92$, Sig.$=0.003$) | Accumulated Population | Transportation Land | $y = 38.563 - 14416.564x$, ($R^2=0.94$, Sig.$=0.002$) |
| Accumulated Population | Cd | $y = 45.414 - 0.661/x$, ($R^2=0.80$, Sig.$=0.016$) | Population Density | III | $y = 37.061 - 1146.586/x$, ($R^2=0.70$, Sig.$=0.037$) |
| Accumulated Population | Pb | $y = -17.376 + 12.821\ln(x)$, ($R^2=0.95$, Sig.$=0.001$) | Accumulated Population Density | Forest | $y = 42.965*x^{0.261}$, ($R^2=0.94$, Sig.$=0.002$) |
| Accumulated Population Density | Al | $y = -87.047 + 16.957\ln(x)$, ($R^2=0.96$, Sig.$=0.001$) | Accumulated Population Density | Grassland | $y = 66.465 - 64.604x$, ($R^2=0.85$, Sig.$=0.009$) |
| Accumulated Population Density | Fe | $y = -88.422 + 15.871\ln(x)$, ($R^2=0.96$, Sig.$=0.001$) | Accumulated Population Density | Transportation Land | $y = 43.393 - 19612.577x$, ($R^2=0.92$, Sig.$=0.003$) |
| Accumulated Population Density | Ni | $y = 1.450 + 20.054\ln(x)$, ($R^2=0.96$, Sig.$=0.001$) | Accumulated GDP | Forest | $y = 90556.374 + 500358.573x$, ($R^2=0.93$, Sig.$=0.002$) |
| Accumulated Population Density | Cu | $y = 52.751 - 65.911/x$, ($R^2=0.87$, Sig.$=0.007$) | Accumulated GDP | Grassland | $y = 565231.417 - 543935.249x$, ($R^2=0.90$, Sig.$=0.004$) |
| Accumulated Population Density | Zn | $y = 50.735 - 232.723/x$, ($R^2=0.88$, Sig.$=0.006$) | Accumulated GDP | Transportation Land | $y = 367785.493 - 16113269.387x$, ($R^2=0.92$, Sig.$=0.002$) |
| Accumulated Population Density | Pb | $y = 14.535 + 17.489\ln(x)$, ($R^2=0.93$, Sig.$=0.002$) | Per Capita GDP | Forest | $y = 7291.924 - 11655.492x$, ($R^2=0.88$, Sig.$=0.006$) |
| GDP | Al | $y = 62402.308 + 7.153x$, ($R^2=0.68$, Sig.$=0.042$) | Per Capita GDP | Grassland | $y = 18459.936 - 12855.251x$, ($R^2=0.87$, Sig.$=0.006$) |
| GDP | Cr | $y = 116487.741 - 623994.562/x$, ($R^2=0.98$, Sig.$=0.000$) | Per Capita GDP | Transportation Land | $y = \exp(9.543 - 376.266x)$, ($R^2=0.91$, Sig.$=0.003$) |
| GDP | Fe | $y = 63101.817 + 3.612x$, ($R^2=0.66$, Sig.$=0.048$) | Accumulated Per Capita GDP | Forest | $y = \exp(9.264 + 2.993x)$, ($R^2=0.86$, Sig.$=0.008$) |
| GDP | Ni | $y = 61830.678 + 1922.278x$, ($R^2=0.70$, Sig.$=0.037$) | Accumulated Per Capita GDP | Grassland | $y = 86235.996 - 89024.133x$, ($R^2=0.92$, Sig.$=0.002$) |
| GDP | Cu | $y = 59807.595 + 2946.703x$, ($R^2=0.83$, Sig.$=0.011$) | Accumulated Per Capita GDP | Transportation Land | $y = \exp(10.956 - 1006.639x)$, ($R^2=0.93$, Sig.$=0.002$) |
| GDP | Pb | $y = 62721.265 + 2843.233x$, ($R^2=0.67$, Sig.$=0.046$) | Accumulated Gross Output Value of Agriculture | Forest | $y = 64329.161 + 259835.845x$, ($R^2=0.92$, Sig.$=0.003$) |
| Accumulated GDP | Al | $y = 61810.012 + 147.475x$, ($R^2=0.98$, Sig.$=0.000$) | Accumulated Gross Output Value of Agriculture | Grassland | $y = 310675.857 - 282212.324x$, ($R^2=0.89$, Sig.$=0.005$) |
| Accumulated GDP | Fe | $y = 74776.315 + 75.15x$, ($R^2=0.97$, Sig.$=0.000$) | Accumulated Gross Output Value of Agriculture | Transportation Land | $y = 209902.565 - 85686748.466x$, ($R^2=0.96$, Sig.$=0.001$) |
| Accumulated GDP | Ni | $y = 22056.352 + 165621.016\ln(x)$, ($R^2=0.97$, Sig.$=0.000$) | CONTAG | Zn | $y = \exp(9.051 - 0.087x)$, ($R^2=0.74$, Sig.$=0.028$) |

**Table 5 (Continuous).** The functional relationships between the economic indices, heavy metal elements and landscape pattern indices

| Indices A (y) | Indices B (x) | Functional Relationships | Indices A (y) | Indices B (x) | Functional Relationships |
|---|---|---|---|---|---|
| Accumulated GDP | Cu | $y = 20939.671 + 189845.505\ln(x)$ , ($R^2= 0.91$, Sig.= 0.003) | PD | Zn | $y = 1.974*1.879^x$ , ($R^2=0.75$ , Sig.= 0.025) |
| Accumulated GDP | Zn | $y = \exp(13.205 − 10.03/x)$ , ($R^2=0.94$, Sig.= 0.001) | DIVISION | Zn | $y = 28.513*x^{1.623}$ , ($R^2= 0.77$, Sig.= 0.022) |
| Accumulated GDP | Cd | $y = \exp(13.282 − 0.038/x)$ , ($R^2=0.83$ , Sig.= 0.012) | CONTAG | Cu | $y = \exp(7.239 − 0.081x)$ , ($R^2=0.70$, Sig.= 0.038) |
| Accumulated GDP | Pb | $y = 129601.73 + 145130.698\ln(x)$ , ($R^2= 0.96$, Sig.= 0.001) | Grassland | Al | $y = 2.143 − 0.228\ln(x)$ , ($R^2= 0.86$, Sig.= 0.008) |
| Per Capita GDP | Al | $y = 1168.75*x^{0.322}$ , ($R^2=0.93$ , Sig.= 0.002) | Grassland | Fe | $y = 2.191 − 0.217\ln(x)$ , ($R^2=0.89$ , Sig.= 0.005) |
| Per Capita GDP | Fe | $y = 1148.026*x^{0.3}$ , ($R^2= 0.93$, Sig.= 0.002) | Grassland | Ni | $y = 0.966 − 0.279\ln(x)$ , ($R^2= 0.91$, Sig.= 0.003) |
| Per Capita GDP | Ni | $y = \exp(9.664 − 1.211/x)$ , ($R^2= 0.94$ , Sig.= 0.001) | Grassland | Cu | $y = 0.966 − 0.319\ln(x)$ , ($R^2= 0.84$ , Sig.= 0.010) |
| Per Capita GDP | Zn | $y = \exp(9.699 − 4.621/x)$ , ($R^2= 0.94$, Sig.= 0.002) | Grassland | Zn | $y = 0.98 − 0.025x$, ($R^2= 0.95$, Sig.= 0.001) |
| Per Capita GDP | Cd, | $y = \exp(9.779 − 0.019/x)$ , ($R^2= 0.98$, Sig.= 0.000) | Grassland | Cd | $y = 1.098\exp(-13.854x)$ , ($R^2= 0.83$, Sig.= 0.012) |
| Per Capita GDP | Pb | $y = \exp(9.626 − 0.591/x)$ , ($R^2= 0.95$, Sig.= 0.001) | Grassland | Pb | $y = 0.957\exp(-0.186x)$ , ($R^2= 0.93$, Sig.= 0.002) |
| Accumulated Per Capita GDP | Al | $y = \exp(11.026 − 520.961/x)$ , ($R^2=0.96$ , Sig.= 0.001) | River | Cu | $y = \exp(-4.723 − 0.128x)$ , ($R^2=0.76$ , Sig.= 0.025) |
| Accumulated Per Capita GDP | Fe | $y = 6106.918 + 12.235x$, ($R^2=0.99$ , Sig.= 0.000) | River | Zn | $y = 0.011 − 0.002\ln(x)$ , ($R^2= 0.68$, Sig.= 0.045) |
| Accumulated Per Capita GDP | Ni | $y = \exp(11.286 − 3.26/x)$ , ($R^2= 0.97$, Sig.= 0.000) | Accumulated Gross Output Value of Agriculture | Fe | $y = -362084.406 + 68803.725\ln(x)$ , ($R^2=0.99$ , Sig.= 0.000) |
| Accumulated Per Capita GDP | Cu | $y = \exp(11.434 − 3.377/x)$ , ($R^2=0.88$ , Sig.= 0.006) | Accumulated Gross Output Value of Agriculture | Ni | $y = 27500.489 + 86966.779\ln(x)$ , ($R^2=0.98$ . Sig.= 0.000) |
| Accumulated Per Capita GDP | Zn | $y = \exp(11.37 − 12.32/x)$ , ($R^2=0.95$ , Sig.= 0.001) | Accumulated Gross Output Value of Agriculture | Cu | $y = 251600.739 − 290115.347/x$, ($R^2=0.92$ , Sig.= 0.003) |
| Accumulated Per Capita GDP | Cd | $y = \exp(11.473 − 0.048/x)$ , ($R^2=0.85$ , Sig.= 0.009) | Accumulated Gross Output Value of Agriculture | Zn | $y = \exp(12.609 − 8.952/x)$ , ($R^2=0.95$, Sig.= 0.001) |
| Accumulated Per Capita GDP | Pb | $y = 15169.098 + 23446.004\ln(x)$ , ($R^2=0.96$ , Sig.= 0.001) | Accumulated Gross Output Value of Agriculture | Cd | $y = \exp(12.679 − 0.034/x)$ , ($R^2=0.83$ , Sig.= 0.011) |
| Accumulated Gross Output Value of Agriculture | Al | $y = -355759.007 + 73460.738\ln(x)$ , ($R^2=0.99$, Sig.= 0.000) | Accumulated Gross Output Value of Agriculture | Pb | $y = 83902.099 + 76300.297\ln(x)$ , ($R^2=0.97$ , Sig.= 0.000) |



**Table 6.** Component matrix

| Indices | Components | | |
|---|---|---|---|
| | 1 | 2 | 3 |
| Accumulated Gross Output Value of Agriculture | .990 | -.014 | .053 |
| Per Capita GDP | .933 | -.323 | -.015 |
| Accumulated Per Capita GDP | .977 | -.086 | .149 |
| Accumulated GDP | .982 | -.040 | .120 |
| GDP | .786 | .332 | .449 |
| Accumulated Population Density | .979 | .052 | .064 |
| Population Density | .156 | .877 | -.304 |
| Accumulated Population | .983 | .005 | .034 |
| Population | -.842 | .366 | .302 |
| River | -.840 | -.342 | .220 |
| Forest | .976 | -.151 | .024 |
| Grassland | -.963 | .195 | -.005 |
| Transportation Land | -.981 | .003 | -.027 |
| SHDI | .763 | -.007 | -.646 |
| IJI | .134 | .947 | .050 |
| PD | .839 | .008 | -.541 |
| CONTAG | -.855 | -.344 | .350 |
| DIVISION | .815 | .082 | -.488 |
| Al | .975 | -.048 | .207 |
| Fe | .969 | -.060 | .220 |
| Ni | .965 | -.010 | .243 |
| Cu | .944 | .200 | .217 |
| Zn | .986 | -.069 | .009 |
| Cd | .838 | -.539 | -.070 |
| Pb | .966 | -.053 | .229 |
| Cr | .783 | .449 | .373 |