# Peer review of "The Potential of Integrating Landscape, Geochemical and 1 Economical Indices to Analyze Watershed Ecological Environment 2 Huan Yua, d, Bo Kongb, Zheng-Wei Hea, Guangxing Wangc, Qing Wangc 3 a College of Earth Sciences, Chengdu Un"

_Hydrology and Earth System Sciences, 2018_

## Referee Comment (RC1) · Anonymous Referee #1 · 22 Oct 2018

Main Comments:

The ecological structure and process are oversimplified, and there is lack of theoretical derivation of the relationship among

different factors affecting the ecological structure and process. How can they be measured? What hypothesis can be made

about the relationship that can lead to empirical test?

How big is your sample size? 9 or 30? Sufficient enough for meaningful statistical analysis?

For regression analysis, the technique is abused by simply regressing one variable on another variable? Any theorectical

model that can guide your model specification? any omitted variable bias? How to deal with colinearity?

Is there any empirical studies in the literature that also attempted to explore the relationship using similar or different

approaches that your study can build on?

Careful english proof reading is needed.

Specific Comments:

On line 32, what is SHDI and IJI? explain them in their first appearance.

On line 33, population and population density are social or socio-economic indicators

Lines 47-50, rephrase the sentence that is unclear.

Lines 50-52, rephrase the sentence that is unclear.

The introduction is too long, with different concepts scattered, lacking of coherence and integration

Section 2.2.4. For landscape pattern metrics, I would suggest to use a table to describe the indices rather than text.

Line 279, population is also considered an indication for measuring economic health and living standard? If this is true,

countries with the highest or lowest population would be most economically health or highest living standards, depending on

relationship mapping?

Line 438-440. You simply did a correlation analysis, how can this statement be supported? Can the correlation between those

indices be sufficient enough to "explain" the regional ecological status and process? the cause-effect relationship? Any

missing variables or factors?

Line 498-500. Not convincing that the ecological status and process of the watershed can be explained by the simple

correlation analysis. See above comments for reason. Lack of a theoretical reasoning and model derivation, simply collecting

some data and doing correlation analysis cannot sufficiently support your conclusion.

Line 506-511. this is obvious, which also explains why you have R2 close to 1. I doubt any value for this correlation

analysis.

In the final paragraph, a question arises regarding the value and theoretical contribution of the study: sufficient enough

for scientific publication or just a class room exercise on correlation analysis?

---

## Referee Comment (RC2) · Anonymous Referee #2 · 31 Oct 2018

This paper attempts to characterize the ecological status of a watershed by relating landscape characteristics, heavy metal concentrations, and economical indices in correlation analyses. While I admire the attempt to treat a river basin as an integrated social-economical-ecological system, and in a transdisciplinary way, this paper has some fundamental problems, which I outline below. 1. There is no theoretical basis for relating the chosen variables to one another. For example: why should heavy metal concentration be related to landscape characteristics, e.g. "CONTAG" and others? 2. Why are heavy metals chosen for this analysis? Nutrient concentrations might be a more logical choice, unless industry or mining is dominant. This however is not explained. 3. The sampling for heavy metals was done at 9 sites, but no information

about sampling frequency or rationale for sampling sites is given. Furthermore, there is an interpolation exercise to derive data for a total 30 points; however, given the size of this basin, I doubt that interpolation is a valid approach. 4. How was autocorrelation among variables accounted for? 5. I miss a description of the main economic activities in the basin, which would allow me to understand the context for the variables chosen. Do people make their living mostly by agricultural means? Or industry? Or..? 6. Use of the word "ecological status" is not defined. Usually this means that some measure or indicators of ecological function are included. (e.g. Biodiversity, intact natural land e.g. parks, nutrient or water retention etc.) How is this used here? 7. Correlation is not causation (e.g. Table 5, use of the phrase "Functional Relationship" means correlation only.) 8. Speaking of Table 5: nearly ever relationship has a p value of <0.05, i.e. is significant. Yet, the discussion suggests others that are not significant. Correlations should be done on the basis of hypothesis testing: Why should certain variables be related? It should not be done in such a way that correlations are developed between every possible combination, as spurious correlations are likely. 9. Finally, work on reducing substantial wordiness. There are many places where the paper is redundant, and contains information that is text-book level. This needs to be reduced, and the higher-level integration, trans-disciplinary literature needs to be presented.

---

## Author Comment (AC1) · 7 Dec 2018

Thanks for reviewers' comments, we have revised our manuscript according to listed comments, and all major changes are red-marked in the revised manuscript. We revised the manuscript by considering all issues mentioned in the reviewers' comments, and we even thoroughly re-wrote the main part of results and discussion.

Please also note the supplement to this comment:
https://www.hydrol-earth-syst-sci-discuss.net/hess-2018-408/hess-2018-408-AC1-supplement.pdf

[Figure]

[Figure]

**Supplement:**

**The Potential of Integrating Landscape, Geochemical and Economical Indices to Analyze Watershed Ecological Environment**

**Huan Yu [a, d], Bo Kong [b], Zheng-Wei He [a], Guangxing Wang [c], Qing Wang [c]**

[a] *College of Earth Sciences, Chengdu University of Technology, 610059, Chengdu, China*

[b] *Institute of Mountain Hazards and Environment, Chinese Academy of Sciences, 610041, Chengdu, China*

[c] *Department of Geography and Environmental Resources, Southern Illinois University, 62901, Carbondale, USA*

[d] *Key Laboratory of Geoscience Spatial Information Technology of Ministry of Land and Resources, Chengdu University of Technology, China*

Correspondence should be addressed to Huan Yu, Email: yuhuan0622@126.com; Telephone: 86-18702846902; Fax: 86-02884075175.

**Abstract:**

A river watershed is a complicated ecosystem, and its spatial structure and temporal dynamics are driven by various natural factors such as soil properties and topographic features, human activities, and their interactions. Thus, characterizing the river watershed ecosystem and monitoring its dynamics is very challenging. In this study, we explored the characteristics of the ecosystem and environment of Yalong River watershed in Ganzi Tibetan Autonomous Prefecture, Sichuan Province of China by analyzing and modeling the relationships among economic indices, heavy metal elements and landscape metrics. Landsat 8 data were used to generate a land cover classification map and to derive landscape pattern indices. Governmental finance statistics yearbook data were referred to provide economic indices. Moreover, a total of 9 water samples (interpolated to 30 samples) were collected from the upstream to the downstream to obtain the values of heavy metal concentrations in the water body. Then, both correlation and regression analyses were applied to analyze and model the relationships among these indices. The results of this study showed that 1) The ecological status and process of this river watershed could be explained by analyzing the relationships among the economic indices, heavy metal elements and landscape pattern indices selected based on correlation analysis; 2) The accumulated economic indices were significantly correlated with Al, Fe and Ni and should be applied to the integrated assessment of the watershed ecological environment; 3) Cu, Zn and Pb were the main elements that showed significant correlations with the forest land; 4) Some landscape patterns indices such as TA and MESH could be used to the integrated assessment of the watershed characteristics because of their strong correlations with the area (or area percentage ) of important landscape types; and 5) transportation land had a close relationship with per capita GDP. 
[revised manuscript text omitted]

Due to a large number of variables of interest (VIs), we first clustered the VIs in terms of their Pearson correlation coefficients (Pearson 1895), and then, the linear regression was performed on those highly correlated VI clusters. In this study, an agglomerative hierarchical clustering analysis (HCA) was used to assess the strength of linear correlation. HCA builds up the clustering hierarchy from bottom to top, i.e., each VI starts with being its own cluster, and the pairs of clusters are then merged as one moves up. The merge happens if the dissimilarity of a pair of clusters is the local minimum. HCA generates a graphical representation – a dendrogram (or tree) – where the VIs are hierarchically grouped together in the hierarchical fashion (e.g., Fig. 10). The height of the dendrogram (tree) implicates the level of dissimilarity, and the process of cluster detection is referred to branch pruning at a desired height (Langfelder et al. 2008). For clustering the strength of linear correlations on VIs, the dissimilarity matric used in the 1st HCA was the determination of Pearson correlation coefficient, $r^2$. The tree was cut at the dissimilarity level of 0.1 ($r^2$=0.9), where the corresponding correlation coefficient would then be larger than 0.949 or smaller than -0.949 in order to maintain the statistical significance.

In addition, stepwise regression was also used to identify the interrelationships among the landscape pattern, heavy metal elements and economic indices, and to determine whether and how the relationships among them could be presented by specific representative factors. The optimal models were assessed based on the coefficient of determination ($R^2$) and statistical significance (Sig.).

**3. Results and discussion**

**3.1 Distribution characteristics of elements in water samples**

Figures 3-6 show the values of water quality parameters for the upper, middle, and lower main channels. The contents of Cr, Ni, Cu, Zn, Cd, and Pb were all below the guideline values for Drinking-water Quality defined by World Health Organization and the Environmental Quality Standards for Surface Water by the Ministry of Environmental Protection of P. R. China. However, the contents of Al and Fe were significantly higher than the guideline values. Spatially, the contents of the elements in the river water generally increased from the upper to downstream. The average values of Al, Fe, Ni, Zn and Pb continuously increased as the water flew to the downstream. The spatial pattern is somewhat alike to that gained in the study of the Fuji River in Japan, in which high-pollution regions were mainly located in the downstream (Shrestha and Kazama 2007). However, the spatial distributions of Cr,

Cu, and Cd values fluctuated from the upstream to the downstream.

**3.2 Analysis of landscape pattern**

Based on the land cover classification results, the landscape pattern indices of the study area were obtained using Fragstats 4.2 software, which are shown in Table 2.

The results indicated that the values of the indices were diversified from the upstream to the downstream except PLADJ, AI, COHESION and MESH. Among the indices,

TA, LPI, CONTAG, PLADJ, COHESION, MESH and AI showed the highest values, while PD, ED, LSI, DIVISION, SPLIT and SHDI had the lowest values in Shiqu county located in the upstream, implying that a health ecological condition was observed in the upstream. In Xinlong county located in the midstream, there were highest values for NP, PD, TE, ED, LSI, DIVISION, SPLIT and SHDI, and lowest values for LPI, PLADJ, IJI and AI, demonstrating that the ecological environment was disturbed and landscape fragmentation was observed. The landscape indices LPI,

PD and DIVISION showed a turning point in the midstream Xinlong County. In

Litang county that had a smallest area, the values of TA, NP, TE, CONTAG,

COHESION and MESH were lowest, while the value of IJI was highest, indicating that the ecological environment needed to be paid attention to.

The AI in all the counties had the values of above 91.5, indicating that the landscape of the study area showed a high degree of aggregation, that is, the ecological environment was still in a good condition. The differences of LPI between the counties were very obvious. All the high values were distributed in the upstream, which meant the large patches dominated the landscape of the region. LSI had the higher values in the downstream, which indicated that the landscape structure was complicated in this region. In addition, by combining the values of CONTAG, PLADJ,

COHESION, DIVISION, MESH, SPLIT and SHDI indices in the table, it was found that the upstream had a better but weaker ecological condition than the downstream.

**3.3 Correlations among landscape pattern, geochemistry and economy indices**

3.3.1 Clustering correlation on VIs

To construct the HCA, a dissimilarity matrix requires to be defined. For the assessment of linear correlation among all the VIs, the dissimilarity was defined as $1-r^2$. The linear dissimilarity matrix was exhibited as a color map in order to visualize the correlation strength (Fig. 9). On the map, the cell color indicates the correlation strength between the pairs of VIs, and as it shifts from blue to red, the linear correlation gets stronger. The HCA for linear correlation resulted a dendrogram shown in Fig. 10, where the red dash line was the threshold at which the tree was pruned. As mentioned before, the threshold was 0.1 in order to maintain the statistical significance. Therefore, only the clusters below 0.1 (red dash line in Fig. 10) were proceeded to further investigation (Table 3).

3.3.2 Regression models of heavy metal elements, landscape pattern and economic indices

Table 3 shows that the members of cluster 1 are chemical elements, the members of cluster 3 and cluster 4 are landscape pattern indices, the members of group 6, 7 and 8 are landscape area statistical indicators, and the members of group 10 are all economic indicators. It is reasonable and easy to understand that the same categories of indicators are clustered in the same group. However, we are concerned about whether there are correlations among different categories of indicators and what quantitative relationships exist between them. Therefore, we explored the regression models between different categories of indices in group 2, group 5 and group 9, respectively. Furthermore, the relationships between geochemical elements and other indicators are what we want to see. Therefore, the regression analysis of each chemical element in group 1 and all other types of indicators was also separately conducted.

The stepwise regression greatly reduced the impacts of multi-collinearity in this study. All the obtained models were significant at the significant level of 0.05 (Table 4). There was only one variable accumulated per capita GDP in the models for the Al element. The variables accumulated per capita GDP and human habitation percentage were involved in the model for the Fe and Ni element. It was found that as the indicators of the economic health in a country and the references for quantifying the intensity of human activities, the economic indices had significantly linear relationships with the Al, Fe and Ni elements at the significant level of 0.05. Furthermore, the accumulated economic indices were involved because of the assembling characteristics of elements from the upstream to the downstream (Cortecci et al. 2009; Li and Zhang 2010; Taylor et al. 2012; Yang et al. 2014; Bu et al. 2016).

The model for the Cu selected the variable forest percentage because of its significant correlation with the element. Two variables forest percentage and transportation land percentage were highly correlated with the Zn element and involved in its prediction model. Cu and Zn are crucial elements for both animals and plants, but they have also been identified as possible specific pollutants in many countries (Comber et al. 2008; Jensen et al. 2016). Many studies have proved that the distribution patterns of Cu and Zn have significant correlations with certain landscape patterns and processes (Stone and Droppo 1996; Lindström 2001; Morse et al. 2016). This was supported by the results of this study.

The variables forest and forest percentage were involved in the model for the Pb element. Forest had a significantly negative correlation with Pb, which is intelligible because forest land has important ecological value and Pb is greatly influenced by human activities. Moreover, the landscape pattern variable TA were sensitive to the land cover area due to the calculation principle and three variables marsh land, marsh land percentage and river that reflect the distribution of landscape area were included in the model. The variables transportation land were involved in the model for the landscape pattern variable MESH. The MESH was used to measure the spatial distribution and degree of landscape fragmentation in former study because it has been proposed as a good single indicator of land division by roads (Jaeger 2000, Li et al. 2010).

A linear relationship between transportation land and per capita GDP could be observed because of both of them involving in the model. Transportation land was highly correlated with the economic index because the traffic condition was one key factor for supporting the local human activities (Gentile and Noekel 2016; Alam et al. 2016). However, no significant correlations were observed between heavy metal elements and landscape pattern indices, which indicates the limitation of analyzing the relationships between the heavy metal elements and the landscape pattern indices to explore the ecological status and process.

**4. Conclusions**

In this study, the relationships between the economic indices, heavy metal elements and landscape pattern indices were explored and used to analyze and characterize the ecosystem and environment of Yalong River watershed within Ganzi Tibetan Autonomous Prefecture, Sichuan Province, using water samples collected in the field and an image derived land cover classification. In summary, this study led to following findings: 1) The ecological status and process of the watershed could be explained by analyzing the relationships among the economic indices, heavy metal elements and landscape pattern indices selected based on correlation analysis; 2) The accumulated economic indices were significantly correlated with Al, Fe and Ni and should be applied to the integrated assessment of the watershed ecological environment. This conforms to the assembling characteristics of the elements in the river from the upstream to the downstream; 3) Cu, Zn and Pb were the main elements that showed significant correlations with the forest land and this was also supported by previous studies; 4) Some landscape patterns indices such as TA and MESH could be used to the integrated assessment of the watershed ecological environment because of their strong correlations with the area (or area percentage ) of important landscape types, i.e., river, marsh and transportation land; however, some limitations of using the landscape pattern indices were also observed, indicating that the selection of the landscape pattern indices was essential; and 5) transportation land had a close relationship with per capita GDP because transportation and mobility were vital constituents of socio-economic development in any country. The conclusions will play a fundamental role in establishing the synthetic models for management of watersheds.

However, it was found that the connection between the economic development and the landscape structure was unable to be established with a simple index analysis. Moreover, analyzing the relationships between the heavy metal elements and the landscape pattern indices to explore the ecological status and process had their limitations. The ecological problems of a watershed could not be revealed through simply analyzing one kind of indices and the sustainable development of the watershed requires an integrated evaluation of hydrologic, ecological and socio-economic factors. A more complicated and comprehensive approach is needed to get an in-depth understanding of the ecological processes and properties of the watershed. Although, at the present, an increasing number of theories and methods for integrated watershed management have been developed, the exploration of quantitative relationships among the driving factors still requires a significant effort and multivariate statistical methods based on sufficient sampling data in the future work could be an alternative.

**Acknowledgments**

This study was supported by the National Natural Science Funds of China (grant no. 41871357), the Sichuan Basic Science and Technology Project (grant no.

18YYJC1148), the Branch of Mountain Sciences, Kathmandu Center for Research and Education, CAS-TU, Chengdu, China (grant no. Y8R3310310), the Hundred

Young Talents Program of the Institute of Mountain Hazards and Environment (grant no. SDSQB-2015-02), the "One-Three-Five" Project of Chinese Academy of Sciences (grant no. SDS-135-1708) and the Science and Technology Service Network Program of the Chinese Academy of Sciences (grant no. Y8R2020022).

[revised manuscript text omitted]

$R^2$, the coefficient of determination; APCGDP, accumulated per capita GDP; HumanhabP, human habitation percentage; ForestP, forest percentage; TransportP, transportation land percentage; Forest, area of forest; Marsh, area of marsh land; MarshP, marsh land percentage; River, area of river; Transport, area of transportation land; TransportP, transportation land percentage; PCGDP, per capita GDP.